# TCR-Directed Therapy in the Treatment of Metastatic Uveal Melanoma

**DOI:** 10.3390/cancers14051215

**Published:** 2022-02-26

**Authors:** Sophia B. Strobel, Devayani Machiraju, Jessica C. Hassel

**Affiliations:** Department of Dermatology, National Center for Tumor Diseases, University Hospital Heidelberg, Code, 69120 Heidelberg, Germany; sophia.strobel@med.uni-heidelberg.de (S.B.S.); devayani.machiraju@med.uni-heidelberg.de (D.M.)

**Keywords:** uveal melanoma, TCR, tebentafusp

## Abstract

**Simple Summary:**

Technical advances in immunotherapies have led to the development of novel T cell receptor (TCR)-based approaches to fight cancer. One of these therapies, tebentafusp, has shown a significant benefit in the overall survival of uveal melanoma patients for the first time. This review focuses on TCR-directed therapies in the treatment of uveal melanoma.

**Abstract:**

Metastatic uveal melanoma (mUM) is one of the most rapidly progressing tumors, with a bad prognosis and no standard-of-care treatment. Immune checkpoint inhibitors have revolutionized cancer therapy and improved overall survival in patients with metastatic cutaneous melanoma (mCM). However, this approach has been largely unimpressive, with no significant impact on the survival of mUM patients. Technical advances in immunotherapies have led to the development of novel T cell receptor (TCR)-based approaches to fight cancer. For the first time in over 50 years, compelling evidence demonstrates the power of TCR-based approaches for survival in mUM patients. Hence, this review summarizes novel TCR-based immunotherapeutic strategies currently in clinical studies for mUM treatment. We also discuss the potential combinational treatments to these strategies to maximize the clinical benefits.

## 1. Introduction

Uveal melanoma (UM) is a rare yet very aggressive tumor arising from melanocytes located in the eye. Early diagnosis is critical, as overall survival depends on the size of the primary tumor. However, most uveal melanomas are diagnosed late, and almost 50% of patients develop metastatic disease, with a limited survival of 6–12 months, with only 8% of patients surviving two years [1,2,3]. Moreover, there is no standard of care for treating metastatic uveal melanoma (mUM), nor has any treatment until recently been shown to improve overall survival in these patients. UM shares a common embryonic origin with cutaneous melanoma (CM); nevertheless, both cancers demonstrate significant differences in their genetic alterations and clinical courses [4]. CM shares a BRAF or NRAS mutation in more than 50% of patients, whereas the most commonly mutated genes in UM are GNA11, GNAQ, BAP1, EIF1AX, and SF3B1. Furthermore, most UM lacks the UV-radiation mutation signature [5,6], and is entailed by a lower tumor mutational burden (TMB) [7,8]. A median somatic mutational load of 1.1 per Mb was observed in UM tumors compared to a median of 18 per Mb in CM tumors. Neoantigens derived from somatic tumor mutations are recognized as foreign to the immune system, thereby conferring immunogenicity to cancer cells. Less TMB may suggest less immunogenicity of UM tumors compared to CM. Thus, metastatic lesions budding from primary tumors of UM may escape immune surveillance, and therapeutic strategies aim to restore the anti-tumor immune response.

Immune checkpoint inhibitors (ICIs) such as anti-programmed death-1 (PD-1) and anti-cytotoxic T-lymphocyte antigen-4 (CTLA-4) inhibit interactions between immune checkpoint proteins and allow T cells to recognize and target cancer cells. ICIs have revolutionized the clinical management of metastatic cutaneous melanoma (mCM), with anti-PD1 monotherapy achieving response rates of approximately 45% and combination therapy (anti-CTLA4 plus anti-PD1) of up to 58% [9,10]. However, the clinical benefit of ICIs has been largely unimpressive in mUM patients, with low response rates, a progression-free survival (PFS) of about 2–3 months, and overall survival (OS) of up to approximately one year [11,12,13]. In general, a high tumor mutational burden and increased expression of immune checkpoint proteins such as PD1 and PDL1 indicate a better response to ICIs in mCM patients [14,15]. However, mUM tumors were shown to have lower TMB and lower expression of PD1 and PDL1 proteins. As the targets for anti-PD1 treatment are less expressed in mUM, this may explain the lack of efficacy of anti-PD1 treatment in these patients. Furthermore, the lower expressions of PD1 and PDL1 suggest that there is either a lack of tumor-specific immune infiltration or that they are suppressed in the TME by other means [16,17,18]. Nevertheless, these data suggest that immunotherapies that not only reverse the exhaustion of existing immune cells but rather drive tumor-specific immune cells to the TME may provide a clinical benefit in mUM patients. The rapidly advancing technologies in the medical field of immune-oncology are leading to the development of novel TCR-based treatments that can drive tumor-specific T cells into TME to fight cancer. 

Functional and tumor-specific immune cells are necessary to identify and kill tumor cells accurately and efficiently. Cytotoxic CD8+ T cells are essential for anti-tumor immunity. Usually, T-cell responses are driven by the T-cell receptor (TCR) and human leukocyte antigen (HLA) interactions. TCRs expressed by CD8+ T-cells employ a glycoprotein α- and β-chain heterodimer that recognizes a tumor peptide in complex with HLA class I molecules on antigen-presenting cells or cancer cells. Upon binding a peptide HLA complex on the cancer cell, a non-covalent oligomeric complex comprising TCR and CD3 signaling molecules initiates signaling activity and enables antigen-specific tumor cell lysis. The strength of the TCR affinity for peptide and HLA complexes determines the activation of lymphocytes. As HLA molecules present both intra- and extracellular tumor proteins, TCRs can be engineered to identify and target tumor antigens that were previously less targetable, suggesting the potential of TCR-based therapies for targeting non-immunogenic cancers such as UM.

Selecting an appropriate tumor antigen is crucial for TCR-based therapies. The targeted antigens presented by HLA on the melanoma cells should not exist or be weakly expressed on normal cells to ensure the safety and effectiveness of TCR-based treatment. Over the past decades, many well-characterized peptide tumor antigens were reported to develop TCR-based therapies. Among these antigen peptides, melanoma antigen recognized by T cells 1 (MART-1), melanoma-associated antigen C2 (MAGE C2), preferentially expressed antigen of melanoma (PRAME), solute carrier family 45 member 2 (SLC45A2), and glycoprotein (gp100) are highly expressed in uveal melanoma tumors and are currently being evaluated in TCR-based clinical trials for the treatment of mUM.

In this review, we provide a summary of novel TCR-based immunotherapeutic strategies, including TCR cell-based (TIL therapy, TCR-engineered T cells) and non-cell-based therapies (ImmTACs) in clinical and preclinical studies for uveal melanoma (Table 1). We also discuss potential combination treatments to these strategies to improve clinical outcomes.

## 2. Cellular-Based TCR Therapy

### 2.1. Genetically Unmodified T Cells

In general, there are two approaches to developing TCR-based cell therapies (Figure 1). Traditionally, therapeutic lymphocytes were produced by expanding tumor-infiltrating lymphocytes (TILs) ex vivo in the presence of immune-stimulating cytokines to enhance their cytotoxic performance [25]. This leads to the production of a T-cell product that resembles a natural TCR, and the autologous TILs are reinfused back to patients without any genetic modifications.

A single-arm, phase II study was conducted to investigate the toxicity and clinical efficacy of the adoptive transfer of TILs in patients with mUM (NCT01814046). In total, 22 patients received lymph-depleting chemotherapy followed by a single infusion of autologous TILs and high-dose of interleukin-2. It is the first report to show the adoptive transfer of autologous TILs to mediate tumor regression in mUM patients [19]. Tumor regression was observed in 35% of patients. Treatment-related toxicities were reported in all patients, which were primarily due to the induction chemotherapy. One treatment-related death due to sepsis and multi-organ failure was reported. Despite the toxicity, these initial results give reason to question the belief that metastatic uveal melanoma is immunotherapy resistant and encourage further investigation of immune-based therapies for this cancer. Accordingly, two other clinical trials are currently investigating the autologous transfer of TILs for mUM patients. The results may provide further information on this strategy’s clinical efficacy and toxicity. However, one of the major limitations of these in vitro expanded T cells is that it is unclear if they have a low affinity to tumor antigens and kill them efficiently. 

### 2.2. Genetically Modified T Cells

A more advanced therapeutic approach of TCR-based cellular therapy features the ex vivo expansion of peripheral blood or tumor-infiltrating T lymphocytes after genetically modifying them by inserting selected genes encoding TCRs that recognize particular tumor antigens with high specificity and affinity and then reinfusing them back to cancer patients. Unlike conventional ACT therapies, the TCR-based treatments rely on tumor antigen presentation by HLA complex; hence, mUM patients with HLA-A*02-positive tumors are eligible to receive genetically modified T cells therapies or non-cellular TCR-based therapy (Figure 2).

MART1-Specific TCR Therapy: MART-1 protein expression could be found in most uveal melanoma tumors [26], and MART-1 reactive T cells can be detected in the peripheral blood 27. Inserting the TCR genes from these T cells into viral vectors and transducing them to T cells led to a potent cytotoxic activity of T cells against MART-1+ and HLA-A*02+ melanoma cell lines with increased cytokine production [27]. Similar results were demonstrated upon modifying TCR genes of previously non-reactive TILs. Together, these data suggest the potential anti-tumor activity of MART-1 TCR therapy for HLA-A*02+ melanoma tumors. Based on the above pre-clinical evidence, a phase I/II clinical trial for MART-TCR therapy (NCT02654821) was initiated, including metastatic melanoma patients. The patients with MART-1 and HLA-A*02:01-positive tumors were recruited into the study. T cells were isolated from the patients, transduced with MART-1 TCR, and expanded ex vivo in the presence of cytokines such as IL7 and IL15. Patients were given chemotherapy prior to the first infusion of MART-1 TCR modified T cells [20]. A total of 12 patients with mUM and mcM were enrolled in the study. Objective responses were seen only in 17% of the patients. However, dose-dependent grade 2 or grade 3 adverse events such as dermatitis, uveitis, cytokine release syndrome, and ototoxicity were reported in all patients. The study has to be terminated due to the severe dose-related toxicity in melanoma patients associated with MART-1 TCR cellular therapy.

PRAME-Specific TCR Therapy: Almost 69% of metastatic uveal melanoma tumors express the cancer-testis antigen PRAME, and given its lack of expression on normal cells, it has been proposed as a therapeutic target for TCR-based therapies in mUM patients [28]. In an experimental study, PRAME-specific T cells reacted against four of seven UM cell lines, and 11 of 16 patient mUM samples were positive for PRAME; 10 of 16 patients expressed HLA class I, and 8 of 16 patients demonstrated coexpression of both PRAME and HLA class I [29]. In addition, in vitro HLA-A*02-restricted, PRAME-specific T cells were able to recognize and react against PRAME-positive uveal melanoma cell lines, suggesting a potential role for PRAME-directed immunotherapy [29,30]. A clinical phase I/II trial is currently ongoing in mUM patients to assess the safety and activity of PRAME-TCR therapy (NCT02743611). Autologous T cells (BPX-701) are modified to target PRAME on melanoma cells and include a biological safety switch that is controllable with rimiducid. Early reports of the project IMA203 (PRAME-TCR therapy) were presented at SITC 2021 [21]. Improved signs of efficacy in sixteen patients with highly refractory solid tumors were reported with a 94% disease control rate and 50% objective responses. The trial was composed of two mUM patients, and one patient (50%) experienced a partial response. The trial is currently investigating higher doses and recruiting patients for a phase Ia dose-escalation cohort.

SLC45A2-Specific TCR Therapy: SLC45A2 is another target currently being investigated in clinical trials for mUM patients. It is localized within the melanosome membrane, and its variants were associated with an increased risk for melanoma [31,32]. Compared to other melanocytic proteins, such as MART-1 and PMEL, the expression of SLC45A2 in normal melanocytes was shown to be less than 2%, resulting in a significantly improved melanoma-to-melanocyte CTL killing index. In vitro, antigen-specific CTLs generated against HLA-A*02:01 and HLA-A*24:02-restricted SLC45A2 peptides were shown to kill most HLA-matched melanoma cells lines, including uveal melanoma cells. In addition, compared to MART-1 and PMEL-specific T cells, which also show a strong reactivity against HLA-A*02+ primary melanocytes along with tumor cells, SLC45A2-specific T cells target tumors cells had a reduced reactivity against HLA-A2+ primary melanocytes [33,34]. Therefore, SLC45A2 specific T cells may selectively target tumor cells with less ability to trigger off target toxicities. Accordingly, a phase I clinical trial including HLA-A*02:01+ and HLA-A*24:02+ mUM patients is currently ongoing to evaluate the clinical efficacy and toxicity of this T cell therapy (NCT03068624). CD8+ SLC45A2-specific T cells are administered through the hepatic artery along with the combination of cyclophosphamide, IL2, and ipilimumab. The study is currently in the recruiting phase and it will be interesting to see the clinical safety and efficacy of this trial.

## 3. Non-Cellular TCR-Based Therapy

Conventional ACT therapies, including TILs or TCR-transduced T cells, are more personalized and laborious. Hence, several alternative therapeutic approaches are being developed to ease the manufacturing issues that come with conventional ACT therapies and broaden the treatment accessibility to patients. In particular, non-cellular-based approaches such as synthetic soluble TCR molecules provide an encouraging alternative to conventional ACT therapies.

Immune-mobilizing monoclonal TCRs against cancer (ImmTAC) molecules are the leading engineered synthetic soluble TCR molecules that are developed to recognize both tumor and T cells. ImmTAC combines a TCR-based targeting system with anti-CD3 antibody fragment effector function [35]. The strong picomolar affinity of engineered TCRs to peptide-HLA results in the coating of tumor cells by ImmTACs. Upon binding, ImmTAC molecules recruit T cells into the TME via interaction between anti-CD3 fragment effector and CD3 and drive selective tumor killing (Figure 3).

Tebentafusp (IMCgp100) is an ImmTAC molecule consisting of an affinity-enhanced TCR fused to an anti-CD3 effector that recognizes a gp100 peptide presented on HLA-A*02:01 with picomolar affinity. Melanoma cells frequently over-express the melanocytic protein gp100. Compared to CM tumors (85%) [36], almost all UM tumors uniformly express Gp100 protein (100%) [26]. The ability of tebentafusp to induce potent anti-tumor activity based on T-cell-mediated killing of exclusively gp100+ and HLA-A*02:01+ melanoma cell lines was demonstrated previously [37,38]. In vitro, the incubation of CD8+ T cells with gp100+ melanoma cell lines demonstrated high lytic activity and cytokine production in the presence of tebentafusp. Tebentafusp-redirected T cells initiate tumor cell death and promote tissue inflammation via the secretion of a wide range of proinflammatory cytokines, including potent chemo-attractants for monocytes [39].

The dose and clinical efficacy of tebentafusp were first investigated (NCT01211262) in HLA-A*02:01+ mUM and mCM patients [22]. The trial reported a response rate of 16% and a 1-year overall survival (OS) rate of 65%, with a surprisingly similar survival and safety profile observed for both mCM and mUM patients. Next, a subsequent phase III randomized clinical study of tebentafusp versus the investigator’s choice (IC) of therapy revealed a significantly better OS (73%) with tebentafusp when compared to IC (59%) [24]. Further, it is important to note that most of the patients in the IC group had received pembrolizumab as the IC therapy. The relative risk of death was reduced by almost half for patients treated with tebentafusp, with a median OS of 21.7 months compared to 16 months with IC. This OS benefit was seen despite a significant but only minorly increased PFS, with 31% of patients being progression-free at six months in the tebentafusp group and 19% in the control group, with a median PFS of 3.3 months compared to 2.9 months. Despite a low response rate of 9% for tebentafusp compared to 5% for IC, patients benefited from the treatment, demonstrating a clear OS benefit for patients with progressive disease as the best response. Like previous clinical studies, treatment-related adverse events in the tebentafusp group were mainly cytokine-mediated and skin-related events, including rash (83%) and pyrexia (76%), were manageable and decreased incidence and severity after the first three to four doses. Notably, no treatment-related deaths were reported in this study [24]. This trial further confirmed the promising clinical activity and safety of tebentafusp in patients with mUM, with survival rates that appear superior to those reported with other existing treatments.

## 4. Challenges and Opportunities

Adoptive cell-based TCR therapies are promising for managing mUM, but there are several challenges ahead, especially considering the personalized nature of the treatments, manufacturing costs, production time limits, and laborious work. In addition to the standardization of manufacturing process, the differences in the duration and frequency of treatment are some of the challenges moving forward. Hence, there is still considerable room for the further improvement of cellular-based therapeutic approaches.

Tebentafusp makes TCR-based immunotherapy accessible for some of the most interesting and highly tumor-specific intracellular antigens and offers pharmacological and manufacturing advantages over cell-based therapies for uveal melanoma. In addition, there seems to be scope for investigating tebentafusp as an adjuvant or neoadjuvant therapy for UM patients whose genetic tumor profile demonstrates a high risk of developing metastases. However, despite the remarkable benefits of tebentafusp, there are also challenges associated with their structure and monoclonal specificity as they may not be effective against tumor variants. Moreover, the two major obstacles commonly associated with the current TCR approaches in general that are essential to conquer are (A) immunosuppressive tumor microenvironment and (B) treatment restricted to HLA-A*02-positive patients.

### 4.1. Immunosuppressive Microenvironment

The persistence of anti-tumor immune responses in the TME is crucial for maximizing the clinical benefits. While TCR-based approaches redirect anti-tumor cytotoxic T cells to the tumor, constant TCR activation and an immunosuppressive tumor environment may eventually result in the exhaustion of tumor-infiltrating immune cells [40,41]. Ideally, to achieve long-lasting efficacy, therapy will need to recruit T cells to tumors and prevent their exhaustion and reactivate dysfunctional tumor-specific T cells. The adjunction of cytokines such as IL-15 and IL-21 has been shown to impact the development, proliferation, differentiation, and survival of T cells and influence the subsequent therapy in vivo during T cell culture. Both cytokines can enhance the expression of granzymes A and B and perforin 1 and improve the cytotoxicity of gp100/HLA-A*02-directed TCR-T cells [42].

In addition, to prevent immune cell exhaustion, there is a good rationale for combining TCR-based approaches with ICIs, which could enhance targeted T cell responses and alleviate the immunosuppressive TME. However, it is important to note that, unlike in CM, immune cells infiltrating UM may have lower expression of PD1, PDL1, and CTLA4 immune checkpoint proteins [17], which may even partially explain the inefficiency of anti-PD1 inhibitors in this tumor entity. In contrast, lymphocyte-activation gene 3 (LAG3) immune checkpoint protein was shown to be highly expressed on tumor-infiltrating CD8+ T cells of high-risk primary UM and was associated with worse survival in UM [17,43]. LAG3 expression was found in 25/43 samples, and the disease-free survival was lower in patients with high expression of LAG3 [44]. Clinical trials to evaluate LAG-3-directed agents (relatlimab) in combination with anti-PD1 (nivolumab) show that relatlimab and nivolumab had long-lasting clinical benefits in mCM patients [45]. Given that the toxicity for the combination was only marginally higher than with nivolumab alone, LAG3 inhibitors are considered to be safe and efficient. Together, these data provide a rationale for combining LAG3 with TCR-based approaches in mUM patients. However, it has to be noted that LAG3 inhibitors demonstrated clinical efficacy when combined with PD1 inhibitors, and the independent activity of LAG3 inhibitors has not been investigated so far. Therefore, to move forward, further studies are required to find the best combinational strategy to combine TCR therapy with LAG3 inhibitors in the presence or absence of PD1 inhibitors. Hence, further studies investigating the potential of cytokines or LAG3 checkpoint inhibitors to overcome the immunosuppressive TME and enhance the efficacy of TCR-based approaches in mUM patients are required.

### 4.2. HLA-Based Selection

A major ongoing limitation of existing TCR-based approaches is the need to restrict enrollment to a special HLA type. HLA-A*02 is the most prevalent HLA I and can be found in approximately 50% of tumors from Caucasian patients, yet it is not as common in other populations. Broadening TCR-based treatments to multiple HLA genotypes and subtypes will increase availability to a broader range of patients who are HLA-A*02-negative. To achieve this, several investigators are developing new TCRs for a broad range of HLA haplotypes to improve the efficacy of TCR therapies [46].

In addition, γδ T cells may provide an alternative platform to αβ T cells and may overcome the hurdles of existing MHC-dependent TCR-based treatment strategies. Unlike αβ T cells, γδ T cells recognize their target cells in an MHC-independent manner. γδ T cells are tissue-resident with innate-like immune responses. γδ T cells have not only been shown to interact with other immune cells but also act like antigen-presenting cells (APCs) and to prime the antigen for αβ T cells [47]. The natural characteristics of γδ T cells enable them to migrate and survive the tumor microenvironment [48]. Furthermore, recent reports have demonstrated the safety and therapeutical application of γδ T cells for cancer treatment [48]. Therefore, identifying a γδ TCR that can recognize tumor cells or developing γδ T cells as a vehicle for αβT cell-derived TCR may lead to an alternative approach for MHC-dependent TCR-based treatments [49]. However, several challenges need to be addressed to move forward with γδ T-cell-based TCR therapy, primarily owing to the difficulty of scaling them to therapeutic numbers ex vivo and their limited presence in the peripheral blood [50]. Researchers are currently investigating the optimum conditions and methods to expand them ex vivo [51]. It will be interesting to see the development and clinical efficiency of γδ T-cell-based TCR therapy for mUM patients irrespective of their HLA status in future studies.

## 5. Conclusions

A deadly prognosis characterizes metastatic uveal melanoma. TCR-based therapies provide many unique advantages over other immunotherapies and show great potential in clinical use, targeting intracellular tumor proteins. Despite the clinical benefits, the extensively laborious process involved in the development of cell-based TCR therapies limits their global accessibility. The soluble TCR-based therapy tebentafusp overcomes the limitation of cell-based T cell therapies and is certainly the leading candidate to make inroads into uveal melanoma survival via T-cell redirection. The challenge moving forward is to consider which treatment combinations will most effectively conquer the obstacles associated with the immunosuppressive microenvironment in tumors that need to be addressed for this treatment to become widely applicable. This requires a more profound knowledge of the tumor–immune system interactions and insight from clinical data and patient outcomes from ongoing clinical investigations. Descriptive studies of responders and nonresponders will be critical in understanding how bispecific TCR molecules benefit and redirect the immune system. It is expected that technologies to generate TCRs and new soluble structures that take advantage of the unique recognition properties of the TCR other than HLA-A*02 positivity will soon result in a significant expansion of these approaches to a broader population of patients with cancer and other diseases.

## Figures and Tables

**Figure 1 cancers-14-01215-f001:**
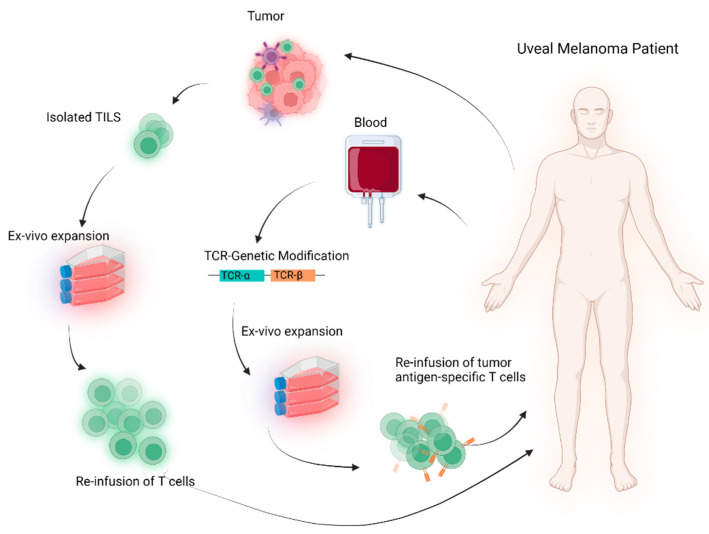
Schematic representation of TCR-based cellular therapies in mUM patients.

**Figure 2 cancers-14-01215-f002:**
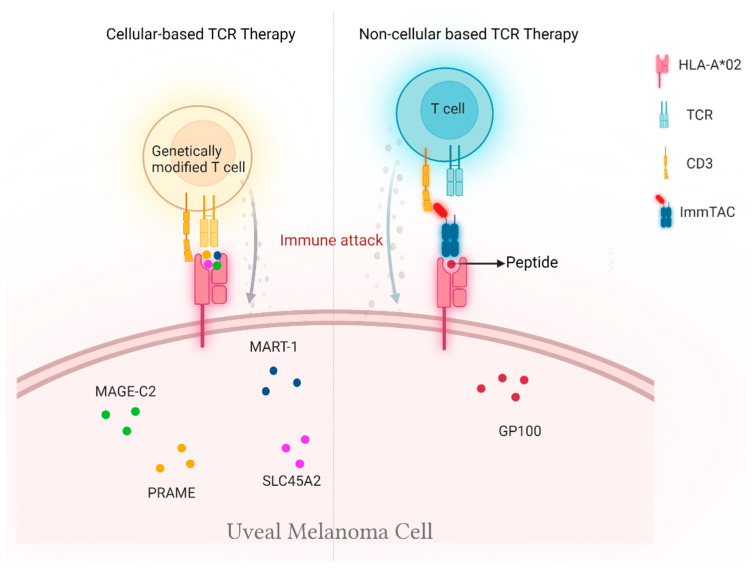
Schematic representation of HLA-A*02-dependent TCR-based cellular therapies in mUM patients.

**Figure 3 cancers-14-01215-f003:**
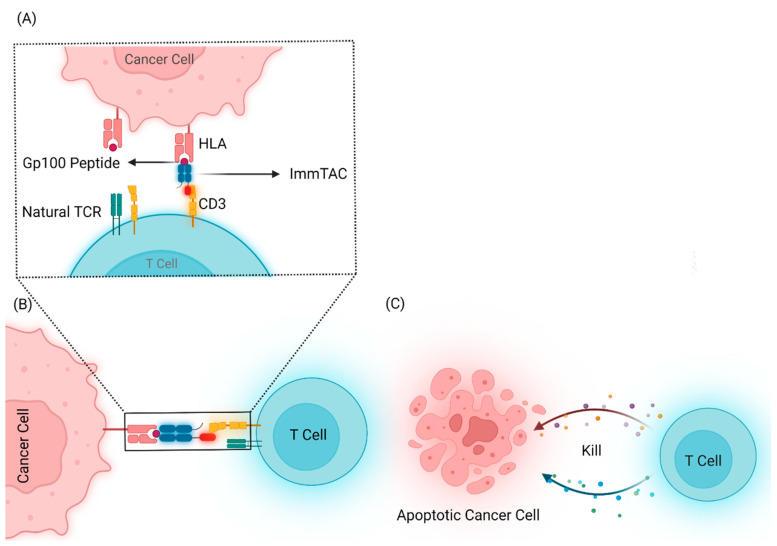
(**A**) ImmTAC molecules are T cell receptor (TCR) x anti-CD3 bispecific fusion proteins that mimic the immune synapse formed by a natural T-cell–cancer cell interaction. (**B**,**C**) Recognition of specific peptide-HLA complexes presented on the cancer cell surface via the affinity-enhanced TCR targeting domain enables recruitment and activation of polyclonal T cells via the CD3-specific effector domain resulting in the targeted release of cytokines and cytolytic mediators to induce cancer cell lysis.

**Table 1 cancers-14-01215-t001:** TCR-based clinical trials in metastatic uveal melanoma patients.

	NCT No.	Phase	Trial	Target	Sponsor/Collaborators	Status	Study Years
HLA-Independent	**Cellular-Based TCR Therapy**						
	Genetically Unmodified T cells						
	NCT01814046	II	Non-myeloablative lymohocyte depletion followed by transfer of autologous TILs with or without high-dose aldesleukin	Non-specific	National Institutes of Health Clinical Center (CC)	Results published [19]	March 2013–July 2017
	NCT03467516	II	Non-myeloablative lymohocyte depletion followed by transfer of autologous TILs with high-dose aldesleukin	Non-Specific	Udai Kammula; University of Pittsburgh	Recruiting (Results pending)	May 2018–December 2023
	NCT04812470	I	Preconditioning chemotherapy with melphlan followed by the transfer of autologous TILs administered via hepatic arterial infusion in addition with IL2	Non-Specific	Sahlgrenska University Hospital; Vastra Gotaland Region; Miltenyi Biomedicine GmbH	Not yet recruiting	November 2021–March 2028
HLA-Dependent	Genetically Modified T cells						
	NCT02654821	I/II	Non-myeloablative lymohocyte depletion followed by transfer of autologous TILs	MART-1	The Netherlands Cancer Institute	Results presented [20]	March 2012–January 2020
	NCT02743611	I/II	Transfer of autologous TCR-engineered T cells in addition to rimiducid	PRAME	Bellicum Pharmaceuticals	Results presented [21]	April 2017–July 2020
	NCT03068624	I	Transfer of autologous TCR-engineered T cells in addition to cyclophosphamide, aldesleukin, ipilimumab	SLC45A2	MD Anderson Cancer Center; NCI	Recruiting (Results pending)	September 2017–September 2021
	NCT04729543	I/II	Pre-treatment with valproic acid and 5′ azacytidine followed by transfer of autologous TCR-engineered T cells	MAGE-C2	Erasmus Medical Center	Recruiting (Results pending)	October 2020-October 2027
	**Non-Cellular TCR-Based Therapy**						
	NCT01211262	I	IMCgp100, a monoclonal T cell receptor anti-CD3 scFv fusion protein	Gp100	Immunocore Ltd.	Completed [22]	September 2010-July 2020
	NCT02570308	I/II	IMCgp100 using the intra-patient escalation dosing regimen	Gp100	Immunocore Ltd.	Results presented [23]	February 2016–January 2021
	NCT03070392	III	Tebentafusp (IMCgp100) versus investigator choice (dacarbazine, ipilimumab, or pembrolizumab) in mUM	Gp100	Immunocore Ltd.	Results published [24]	October 2017–March 2023

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
