# Peer review of "TCR-Directed Therapy in the Treatment of Metastatic Uveal Melanoma"

_cancers, 2022, doi:10.3390/cancers14051215_

Round 1
Reviewer 1 Report
I´d like to thank the authors for the opportunity to review this comprehensive an timely short review on the developing field of TCR based therapies for patients with metastatic uveal melanoma.
I do only have minor comments:
line 80: “There are mainly 3 types of tumor antigens” stands alone – and no further explanation follows did the authors originally plan to name the three types? Please add either details about the three types or delete.
line 83: As Lag-3 blocking antibodies did only demonstrate activity in combination with PD-1 antibodies and not as a single agent could the authors discuss their rationale for a double combination with “only” a LAG-3 antibody over a triple combination with a PD-1 blocking agent.
In addition a few suggestions on typos/corrections:
line 15/Abstract impact instead of impacts
line 21/Abstract benefit instead of benefits
line 28/29 introduction: …survival of 6-12 months and only 8% of patients surviving two years.
line 30: …overall survival in these patients
line 33: course instead of courses
line 56: absence of a type 1 immune response
line 308: …only marginally higher than with nivolumab alone.
line 339: …. for obtaining clinically efficient…..
line 358: … significant expansion of these….
throughout the review – the authors switch frequently from present to past tense, sometimes even within the same sentence. Please correct and use either present or past tense.
Author Response
Reviewer#1
Open Review
Comments and Suggestions for Authors
I'd like to thank the authors for the opportunity to review this comprehensive an timely short review on the developing field of TCR based therapies for patients with metastatic uveal melanoma.
Author's Reply: We want to thank Reviewer#1 for going through the manuscript and providing valuable feedback.
I do only have minor comments:
- line 80: “There are mainly 3 types of tumor antigens" stands alone – and no further explanation follows did the authors originally plan to name the three types? Please add either details about the three types or delete.
Author's Reply: We have removed the sentence “There are mainly 3 types of tumor antigens” (Line:80).
- line 83: As Lag-3 blocking antibodies did only demonstrate activity in combination with PD-1 antibodies and not as a single agent could the authors discuss their rationale for a double combination with “only” a LAG-3 antibody over a triple combination with a PD-1 blocking agent.
Author's Reply: We thank Reviewer#1 for bringing this subject, and we incorporated the information on the LAG3 combinational strategy in the revised manuscript (Line:306-311).
- In addition a few suggestions on typos/corrections:
- line 15/Abstract impact instead of impactsCo
- line 21/Abstract benefit instead of benefits
- line 28/29 introduction: …survival of 6-12 months and only 8% of patients survivingtwo years.
- line 30: …overall survival in thesepatients
- line 33: course instead of courses
- line 56: absence of atype 1 immune response
- line 308: …only marginally higher than with nivolumab alone.
- line 339: …. for obtaining clinicallyefficient…..
- line 358: … significant expansion ofthese….
- throughout the review – the authors switch frequently from present to past tense, sometimes even within the same sentence. Please correct and use either present or past tense.
Author's Reply: We corrected all the above-mentioned typing errors (comment#3), and in addition, the revised manuscript has gone through the English edition. So, we hope that it is okay.
Reviewer 2 Report
Authors should re design the appropriate table via adding current all TCR based clinical trials in metastatic uveal melanoma patients.
The promising part of this paper is inclusion of HLA based strategies extensively. For this reason, it presents a wide variety of application types in a well organized way.
HLA related studies could be shown in a summary graphics.
Author Response
Reviewer#2
Author’s Reply: We would like to thank Reviewer#2 for the suggestions and driving us to revise our manuscript for a better version.
Comments and Suggestions for Authors
- Authors should re-design the appropriate table via adding current all TCR based clinical trials in metastatic uveal melanoma patients.
Author’s Reply: We have re-designed the table.1 and also included a new clinical trial that is yet to be started for mUM patients (NCT04812470) according to the Reviewer suggestion. We have listed all the TCR-based clinical trials for uveal melanoma that were completed or ongoing as of Nov 2021(Source: clinical trials.gov.in). In total, we found 184 clinical trials that are either terminated or currently ongoing for uveal melanoma. Among them, we only found seven clinical trials that were based on TCR or TILs. In addition, there were other ACT therapies, such as CAR-T cell therapy. However, the study focused on TCR-based treatment and so we did not included CAR T cell therapies in table 1. Hope that’s fine.
- The promising part of this paper is inclusion of HLA based strategies extensively. For this reason, it presents a wide variety of application types in a well-organized way. HLA related studies could be shown in a summary graphics.
Author’s Reply: We thank Reviewer#2 for the suggestion. We tried to conclude the HLA-based TCR therapy in summary graphics and incorporated it in the revised manuscript (Figure 2). We hope that adds to the vision.
Reviewer 3 Report
TCR Directed Therapy in the Treatment of Metastatic Uveal Melanoma
Sophia B. Strobel, Devayani Machiraju, and Jessica C. Hassel
Overview:
This manuscript is a short review on the immunotherapeutic approaches against metastatic uveal melanoma (mUM) in particular and metastatic cutaneous melanoma (mCM) in general. Authors have described the cellular and non-cellular, TCR-based immunotherapeutic approaches, their individual efficiencies, shortcomings, and the scope of further improvements. The review is well written, concise, with a clear focus on the topic. I recommend its publication in the journal “MDPI Cancers” after addressing few minor concerns I have listed below.
Specific Comments:
- Lines 31-38. Please add more details about the mutational burdens of both the types mUM and mCM and rationalize it with lower immunogenicity of the mUM. Authors have provided appropriate references, however, considering the central focus of the manuscript, a detailed rationale will improve the quality further.
- At some places, authors are too specific about the clinical trials, number of patients, responders, and non-responders. For example, line 156-168. I suggest replacing such details with the statistics on the overall response.
- At some places, the sentences are too long, mostly due to abbreviations. For example, line 211-213, line 250-253. Please check the whole manuscript for such sentences and shorten them in some manner.
- Quality of the figures is very poor. In addition to better resolution, please increase the font size of the text on the figures so that it is readable.
Author Response
Reviewer#3
Open Review
Comments and Suggestions for Authors
TCR Directed Therapy in the Treatment of Metastatic Uveal Melanoma
Sophia B. Strobel, Devayani Machiraju, and Jessica C. Hassel
Overview:
This manuscript is a short review on the immunotherapeutic approaches against metastatic uveal melanoma (mUM) in particular and metastatic cutaneous melanoma (mCM) in general. Authors have described the cellular and non-cellular, TCR-based immunotherapeutic approaches, their individual efficiencies, shortcomings, and the scope of further improvements. The review is well written, concise, with a clear focus on the topic. I recommend its publication in the journal “MDPI Cancers” after addressing few minor concerns I have listed below.
Author’s Reply: We would like to thank Reviewer#3 for taking time to go through the Review and for providing very valuable feedback.
Specific Comments:
- Lines 31-38. Please add more details about the mutational burdens of both the types mUM and mCM and rationalize it with lower immunogenicity of the mUM. Authors have provided appropriate references, however, considering the central focus of the manuscript, a detailed rationale will improve the quality further.
Author’s Reply: We detailed the reason for low immunogenicity of mUM in the revised manuscript (Line: 35-41).
- At some places, authors are too specific about the clinical trials, number of patients, responders, and non-responders. For example, line 156-168. I suggest replacing such details with the statistics on the overall response.
Author’s Reply: We replaced the absolute numbers with percentage of overall responses in the revised manuscript (Line: 161-16; 180-183;240-252).
- At some places, the sentences are too long, mostly due to abbreviations. For example, line 211-213, line 250-253. Please check the whole manuscript for such sentences and shorten them in some manner.
Author’s Reply: We tried to revise the manuscript for such sentences and shorten them down. In addition, we revised manuscript with the help of a native speaker.
- Quality of the figures is very poor. In addition to better resolution, please increase the font size of the text on the figures so that it is readable.
Author’s Reply: We have re-designed the figures in the revised manuscript for a better quality and hope that's fine.
Reviewer 4 Report
TCR Directed Therapy in the Treatment of Metastatic Uveal Melanoma. Strobel et al.
This is a very nice overview of existing data and ongoing clinical studies on TcR driven strategies for treatment of metastatic uveal melanoma. The topic is of renewed relevance based on the recent approval by FDA of the TcR directed drug Tebentafusp for uveal melanoma.
Comments
The manuscript would benefit from language revision. E.g. the start of the sentence on page 2, line 73 do not make sense “As both intra- and extracellular tumor proteins are presented by MHC molecules, TCR-bearing T cells can penetrate tumors more efficiently…..”.
Also, the sentence page 2, line 80 “There are mainly three types of tumor antigens.” does not connect to the following text.
Table 1 could be more systematical built to ease the overview e.g. sub-grouping according to cell-based/non-cell based, Ag target, and trial phase.
Page 4, line 150 please explain restriction to HLA-A0201
Page 5, line 182, “revealed improved signs of efficacy….” Please revise the sentence so it is clear for the reader how many patients were treated and how did uM patients respond. Also, the HLA restriction should be mentioned.
Page 5, line 205 “Such laborious and technical processes have proven to be difficult to translate into a widely available therapy” Please consider the fact that CART is approved SOC with broad availability.
Page 7, 4. Challenges and Opportunities. Please also consider the differences in duration and frequency of the treatments as well as the economic perspectives.
Author Response
Reviewer#4
Open Review
Comments and Suggestions for Authors
TCR Directed Therapy in the Treatment of Metastatic Uveal Melanoma. Strobel et al.
This is a very nice overview of existing data and ongoing clinical studies on TcR driven strategies for treatment of metastatic uveal melanoma. The topic is of renewed relevance based on the recent approval by FDA of the TcR directed drug Tebentafusp for uveal melanoma.
Author’s Reply: We would like to thank Reviewer#3 for the valuable suggestions. We truly appreciate the feedback.
Comments
- The manuscript would benefit from language revision. E.g. the start of the sentence on page 2, line 73 do not make sense “As both intra- and extracellular tumor proteins are presented by MHC molecules, TCR-bearing T cells can penetrate tumors more efficiently…..”.
Author’s Reply: We apologize for the mistake. We have revised the text, and hope the sentence is clear to understand (Line 73-76).
- Also, the sentence page 2, line 80 “There are mainly three types of tumor antigens.” does not connect to the following text.
Author’s Reply: In the revised manuscript, we have removed the sentence (Line:80).
- Table 1 could be more systematical built to ease the overview e.g. sub-grouping according to cell-based/non-cell based, Ag target, and trial phase.
Author’s Reply: Table 1. has been re-designed in the revised manuscript. We hope that's fine.
- Page 4, line 150 please explain restriction to HLA-A0201
Author’s Reply: The restriction HLA-A*02 has now been described in the line (Line:145-147; 154)
- Page 5, line 182, “revealed improved signs of efficacy….” Please revise the sentence so it is clear for the reader how many patients were treated and how did uM patients respond. Also, the HLA restriction should be mentioned.
Author’s Reply: We incorporated more details on the trial results in the revised manuscript (Line: 180-185).
- Page 5, line 205 “Such laborious and technical processes have proven to be difficult to translate into a widely available therapy” Please consider the fact that CART is approved SOC with broad availability.
Author’s Reply: We thank you for the correction; in the revised manuscript, we rewrote the sentence (Line:204-206).
- Page 7, 4. Challenges and Opportunities. Please also consider the differences in duration and frequency of the treatments as well as the economic perspectives.
Author’s Reply: We extended the limitations and included the suggested issues in the revised manuscript (Line:264-269).